# Enhanced Recovery of Bioactive Compounds from Cagaita and Mamacadela Fruits Using Natural Deep Eutectic Solvents (NADES) and Ethanol: A Comparative Study

**DOI:** 10.3390/plants14162596

**Published:** 2025-08-21

**Authors:** Jaqueline Ferreira Silva, Carmen Torres Guedes, Eloize da Silva Alves, Évelin Lemos de Oliveira, Eduardo Cesar Meurer, Suelen Siqueira dos Santos, Mônica Regina da Silva Scapim, Grasiele Scaramal Madrona

**Affiliations:** 1Postgraduate in Food Sciences, State University of Maringá, Maringá 87020-000, Paraná, Brazil; jaquelinesferreirasilva@gmail.com (J.F.S.); eloizeetaus@gmail.com (E.d.S.A.); 2Postgraduate in Food Engineering, State University of Maringá, Maringá 87020-000, Paraná, Brazil; ctorresguedes@gmail.com; 3Department of Chemistry, State University of Maringá, Maringá 87020-000, Paraná, Brazil; elemosoliveira01@gmail.com; 4Food Engineering, Federal University Paraná, Jandaia do Sul 86900-000, Paraná, Brazil; eduardo.meurer@ufpr.br (E.C.M.); suelensiqueira@ufpr.br (S.S.d.S.); 5Food Engineering Departament, State University of Maringá, Maringá 87020-000, Paraná, Brazil; mrsscapim@uem.br

**Keywords:** *Eugenia dysenterica* DC, *Brosimum Gaudichaudii* Trécul, extraction, antioxidant, green solvents, native fruits, Cerrado

## Abstract

The native fruits of the Cerrado have an interesting composition of bioactive compounds responsible for antioxidant, anti-inflammatory, and antimicrobial activities, with technological potential for functional industries. This study investigated the extraction of bioactive compounds in cagaita and mamacadela fruits, under different conditions, using eutectic solvents based on choline chloride (CC) with citric acid (CA) or tartaric acid (TA), plus ethanol as reference. For a better understanding of the extracts, their antioxidant capacity was assessed by the DPPH^•^, FRAP, ABTS^•+^, and total phenolic compounds and flavonoids assays, as well as for color, water activity, and identification of bioactive compounds by mass spectrometry. Additionally, the carotenoid contents were evaluated in the ethanolic extracts. The results showed that ethanol was efficient for the extraction of flavonoids and presented advantages demonstrated in the antioxidant analyses of ABTS^•+^ and FRAP. However, eutectic solvents stood out in the extraction of phenolic compounds, with yields 14.0 and 4.5 times higher than ethanol for mamacadela and cagaita, respectively. In addition, when compared to cagaita, mamacadela had twice the carotenoid content. Furthermore, the CC:TA solvent was the most efficient, demonstrating, by DI-ESI-MS, 29 phenolic compounds in mamacadela and 27 in cagaita. Therefore, the extracts obtained present potential for use as natural pigments, adding value to the fruits and encouraging their exploration by industries.

## 1. Introduction

The Cerrado, one of the main Brazilian biomes, is a region of great ecological, social, and economic importance. It is the second-largest biome in South America, covering about 23.3% of Brazilian territory, or approximately 1,983,017 km^2^ [1]. The biodiversity of the Cerrado is remarkable, and it is considered a global hotspot. This biome is home to more than 12,000 plant species [2], and accounts for 5% of the world’s plant and animal species.

Several of its fruit and tree species are potentially interesting for agroindustry, but they are still unknown, little explored, and/or underutilized [3,4]. In addition to its rich flora, the Cerrado plays a crucial role in the hydrological cycle of Brazil [5], further increasing the importance of this biome. Due to the current rates of Cerrado deforestation, about 60% to 80% of the typical flora in the Cerrado region may be at risk of extinction [6,7].

The rapid expansion of agribusiness, poorly developed and inadequately enforced environmental laws, and declining funding for research and conservation are driving a major biodiversity collapse. Transportation infrastructure is a leading contributor to habitat loss, posing a significant threat to mammal species [8,9,10].

Research on native fruits promotes environmental preservation by offering a sustainable alternative to crops like soy, as these fruits are part of the local flora and support biome conservation and commercialization [11,12].

One of the strategies used in this type of research is the extraction of bioactive compounds with different solvents. The objective is to identify the solvent that is most effective in extracting, quantifying, and identifying these compounds in fruits. The choice of solvent is critical, as the use of each one results in different chemical properties, yield, molecular weight, selectivity, and bioactivity [13,14].

Given this, deep eutectic solvents (DESs) are receiving increasing attention from researchers across various areas of science and technology because of the environmentally friendly, biodegradable, and highly soluble nature of these solvents. Beyond their wide range of applications, particularly those associated with introducing new functional groups and surface modifications, one of the major challenges associated with their use is their high viscosity. This can be reduced by incorporating water into the solvent system, applying heat, or using large volumes of DESs. These solvents are described as simple, eco-friendly, cost-effective, and easy to obtain [15,16,17,18,19].

Ethanol is an organic solvent with low toxicity, and which is widely recognized and used for the extraction of bioactive compounds, mainly due to its lower environmental impact and toxicity compared to methanol [20,21]. However, eutectic solvents have emerged as a safer option compared to conventional organic solvents such as ethanol, as compared to the previous approach, in which only the good performance in the extraction of bioactive compounds was considered in evaluating an organic solvent, even if it was considered to demonstrate lower sustainability [22].

Comparative studies on solvent efficiency for the purposes of extraction allow the industry to select the most appropriate option based on the target compounds and intended applications. Thus, this work aims (1) to extract bioactive compounds from two native fruits of the Cerrado (mamacadela and cagaita) under different conditions and utilizing different solvents (eutectic and organic); (2) to evaluate the color and the activity in water and quantify the bioactive compounds; and (3) to identify by mass spectrometry (DI-ESI-MS) the compounds present in the different extracts.

## 2. Results

### 2.1. Coloration and Water Activity of Extracts

The color analysis and water activity (AW) tests were performed in the different extracts studied, and the values were compared among themselves, as shown in Table 1.

The color characterization was explored in this study through extract characterization. The use of different extraction solvents allows the industry to make the best choice according to its needs.

The processes utilizing different solvents resulted in different colors of extracts, as can be seen in Figure 1. The ethanolic extracts mainly showed an intense yellow color, compared to eutectic solvents for both fruits.

### 2.2. Bioactive Compounds of Extracts

The extracts of mamacadela and cagaita obtained by different processes and solvents were evaluated for antioxidant activity and their bioactive compounds, as presented in Table 2.

The different solvents exhibited significant differences in antioxidant values, including those for total phenolic compounds (TPC), flavonoids, ABTS^•+^, FRAP, and DPPH^•^. This variation was also observed between the different fruits. In general, eutectic solvents facilitated the extraction of bioactive compounds (DPPH^•^, FRAP, and TPC), resulting in higher values (*p* < 0.05), while ethanol stood out in the flavonoids and ABTS assays.

In addition, between the fruits, the cagaita presented significant differences (*p* < 0.05) for all other bioactive compound analyses (TPC, ABTS^•+^ and DPPH^•^), mainly when using ethanol as an extractor solvent, when compared to mamacadela, except for flavonoids, for which the mamacadela had higher values.

The total carotenoids are described in Figure 2, in which the mamacadela has the highest values in cases with the application of agitation, with those values being two times higher than those recorded for cagaita.

### 2.3. Mass Spectrometry of the Different Extracts

The results of mass spectrometry analysis for extracts of mamacadela fruit are described in Table 3.

The eutectic solvent choline chloride with tartaric acid (CC:TA) stood out in this test, given the greater identification of bioactive compounds, when compared with other solvents. The cagaita fruit is presented in Table 4; the same solvent obtained the highest levels of relevant compounds present in the extracts.

## 3. Discussion

### 3.1. Coloration and Water Activity of Extracts

The CIELAB definitions describe the L* representing the difference in brightness, in which positive values represent a tendency towards brighter colors and negative values towards darker colors. A positive value associated with a* indicates a tendency to red, and a negative value trends towards green. A positive value of b* indicates a tendency towards yellow, while negative values of b* indicate a tendency towards blue [23]. The colorations of the extracts are shown in Table 1. It is possible to observe that, considering all samples, there are no significant differences (*p* < 0.05) for the parameter L*, except in extracts using ethanol and conventional bath. Regardless of the temperature variation, the same behavior was observed when comparing the different fruits.

The agitation bath treatment resulted in higher L*, a*, and b* values for the mamacadela fruit, indicating a significant shift toward brighter colors (L*), red hues (+a*), and yellow tones (+b*). The Wi parameter increased notably when eutectic solvents were used, for both fruits, with the rise in whiteness aligning with the L* values, as also observed in Figure 1. Regarding Yi, ethanol-based extracts (agitated at 30 or 60 °C, or conventionally extracted at 60 °C) exhibited higher yellowing index values, demonstrating the effectiveness of this method in enhancing the yellowish coloration, which is likely associated with the carotenoid contents of the fruits. This effect was particularly pronounced in cagaita extracts, in which Yi values were higher than those of mamacadela fruit.

The water activity associated with eutectic solvent treatments was significantly lower than that associated with ethanol treatments. This effect may be associated with the high viscosity of eutectic solvents, which is one of the challenges in and limitations of the use of this type of solvent, in addition to matrix effects and compatibility issues relative to the analytical instrumentation [24]. However, the water content mixed with the eutectic solvent should not be too high, as it may break the hydrogen bonds within the constituents of the DESs [25].

### 3.2. Bioactive Compounds of Extracts

Phenolic compounds were significantly better-extracted by eutectic solvents, demonstrating the latter’s effectiveness as an alternative for the extraction of antioxidant compounds. Alasalvar [20] also used eutectic solvents to extract phenolic compounds from imortelle flowers (*Helichrysum italicum*), obtaining a high content of phenolic compounds (101.5 mg gallic acid equivalent (GAE)/100 g) when using lactic acid and fructose as solvents.

In the present study, eutectic solvents yielded even higher contents, reaching 144.21 ± 6.37 mg GAE/100 g and 141.91 ± 21.15 mg GAE/100 g in mamacadela extracts with CC:TA and choline chloride with citric acid (CC:CA), respectively, and 171.32 ± 21.52 mg GAE/100 g in the cagaita extract with CC:CA. Other studies have shown that choline chloride–citric acid outperforms conventional solvents in polyphenol extraction efficiency when used in the context of microalgae biomass, proving to be a sustainable and environmentally friendly alternative [19,26]. Soukaina et al. [27] also demonstrated that phenolic acids have a greater affinity for eutectic solvents and exhibit increased selectivity towards flavonoids.

The different fruits were evaluated using several quantification methods, with the mamacadela ethanol extract at 60 °C with agitation yielding the highest results across various methodologies (ABTS^•+^ 190.88 20.83 mg Trolox/100 g, FRAP 153.32 7.47 mg Trolox/100 g, and flavonoids 135.18 3.98 μg eq Quercetin/100 g).

The eutectic solvents were the only ones that enabled detection by the DPPH^•^ method for mamacadela. Determining the variations in the quantification of antioxidant and compound capacity is crucial for providing a comprehensive profile of the compounds present in the extracts.

The total carotenoids (Figure 2) for the different ethanol extraction treatments showed a significant difference relative to the extraction equipment used, with the agitated bath yielding results twice as high as the conventional bath. The various extraction temperatures did not show significant differences in their results.

Mamacadela also demonstrated superior carotenoid contents compared to cagaita, with much higher total carotenoid values (Figure 2). These findings indicate that mamacadela is a promising source of carotenoids, either by extraction for potential use as an ingredient in human food, or in its natural form.

Therefore, the cagaita extract showed higher values with ethanol extraction, agitation, and a temperature of 60 °C. Increasing the temperature improved extraction for some samples, realizing enhanced solubility and diffusion coefficients, and thereby optimizing the extraction process [27,28,29].

However, high temperatures in certain plant samples can damage the cell walls and subcellular components, leading to the release of large amounts of active compounds and enhancing the ability to capture free radicals. As a result, higher temperatures cause tissue softening and the release of bound antioxidant components [29].

### 3.3. Mass Spectrometry of Different Extracts

Mass spectrometry analyzes the chemical structures of bioactive compounds, aiding in their characterization and classification [30], and facilitating the identification of phenolic compounds in the different extracts obtained. Table 3 lists the identified phenolic compounds, including caffeic acid, gallic acid, syringic acid, enterodiol, *p*-coumaric acid, *p*-hydroxybenzoic acid, propocatechuic acid, trans-cinnamic acid, chlorogenic acid, rosmarinic acid, salvianolic acid, and ferulic acid.

Several flavonoids and their derivatives were also detected, including quercetin, myricetin, eriodictyol, hesperetin, kaempferol, luteolin, isorhamnetin, naringenin, and apigenin. Additionally, some anthocyanins, such as cyanidin (cyanidin-3-O-arabinoside, cyanidin-3-O-glucoside, and cyanidin-3,5-O-dihexiside), were identified, as outlined in this study [31,32,33,34,35,36,37,38,39].

Twenty-nine compounds were identified using the CC:TA solvent, including fifteen phenolic compounds, twelve flavonoids, and two anthocyanins, making it the extract with the highest number of compounds identified and the method associated with their greatest intensity. The extracts with CC:CA and ethanol at 30 °C CB, 60 °C CB, 30 °C BA, and 60 °C BA identified 19, 13, 16, 15, and 17 phenolics, respectively. The mamacadela extract contained four compounds that were not identified in the other extracts (eriodictyol-O-rutinoside, hesperetin, cyanidin-3-O-glucoside, and vanillic acid).

The eriodictyol compounds found in the fruit analyzed represent a flavonoid that belongs to a subclass of flavanones that have therapeutic action in neuroprotection, cardioprotective activity, hepatoprotective activity, antidiabetes and obesity, and skin protection and that have highly analgesic, antioxidant, and anti-inflammatory effects, in adition to antipyretic and antinociceptive actions, antitumor activity, and more [33].

Hesperetin is a flavonoid compound that exhibits several pharmacological effects, such as anti-inflammatory, antitumor, antioxidant, anti-aging, and neuroprotective properties [37]; in addition, there are studies suggesting favorable correlations in the treatment of breast cancer [40].

The other compound identified in mamacadela extracts was cyanidine-3-O-glycoside, which is an anthocyanin that has antioxidant and anti-inflammatory properties; in the gastrointestinal tract, it can produce bioactive phenolic metabolites such as protocatechic acid, phloroglucinaldehyde, vanillic acid, and ferulic acid, which increase the bioavailability of cyanidine-3-O-glycoside and contribute to the mucous barrier and microbiota [38].

Vanillic acid is an aromatic acid that demonstrates a potent therapeutic activity relative to anticancer, inhibiting the onset and progression of various malignant tumors and in the treatment of inflammation [35,41].

The cagaita extracts showed a higher number of phenolic compounds, as identified in the eutectic solvent CC:TA sample (twenty-seven compounds, including twelve phenolic acids, fourteen flavonoids, and one anthocyanin); this was also the best solvent for mamacadela.

The other extracts had a higher efficiency of extraction of phenolics when compared with the extracts of mamacadela; for ethanol, 23 compounds were identified for CC:CA, 25 for 30 °C CB, 19 for 60 °C CB, 22 for 30 °C BA, and 22 for 60 °C BA (Table 4). The cagaita extracts contained four phenolic compounds that were not identified in the mamacadela extracts: eriodictyol-O-hexoside, kaempferol-hexoside, luteolin, and ferulic acid.

It is important to highlight that one of the most well-known and studied natural flavonoids is kaempferol, which was identified in cagaita; it has anticarcinogenic and anti-inflammatory effects and exhibits antibacterial, antifungal, and antiprotozoal activities [36,42,43,44].

Ferrulic acid is a natural phenolic phytochemical utilized in antioxidant, anti-inflammatory, antimicrobial, antiallergic, hepatoprotective, anticarcinogenic, and antithrombotic applications [45,46,47].

Another compound found in this work is luteolin, which is a type of flavonoid that has antioxidant, antitumor, and anti-inflammatory properties, as well as a very significant potential for use as a chemopreventive dietary molecule, inhibiting tumor cell metastasis and angiogenesis. The recent relevant scientific literature has reported cardiac protective effects [33,48,49].

The screening (which is used for both conditions) in this study aims to confirm the presence of phenolic compounds in a sample, usually without a reference standard, but with preliminary information on the exact mass and isotopic pattern of the expected molecular formula or adducts; this allows the detection of known and unknown compounds in various food source matrices [50].

Soukaina et al. [27] used eutectic solvents, choline chloride–lactic acid and choline chloride–glycerol (1:2), in their studies and observed that phenolic acids had a strong affinity for these solvents. This finding is particularly relevant to our study, especially regarding the extracts of mamacadela. On the other hand, our results suggest that the proposed eutectic solvents show a greater selectivity for flavonoids, such as derivatives of apigenin, kaempferol, and quercetin, as well as some cyanidin derivatives. In this study, the eutectic solvent CC:TA demonstrated a greater affinity for phenolic compounds, leading to better results.

The method studied, using DES, in addition to demonstrating sustainability, proved to be efficient in extracting bioactive compounds from natural sources, thus highlighting the importance of this research in the area of natural product chemistry [51].

However, one of the major difficulties in using eutectic solvents is the difficulty of separating the extracted compounds; this is the greatest challenge in using this type of solvent [52]. Therefore, studies like this are important for the dissemination of knowledge in the food, cosmetic, and pharmaceutical industries, given that demand in the use of natural products has increased greatly in recent years, and several studies have already demonstrated the effectiveness of natural pigments as dyes [53].

## 4. Materials and Methods

### 4.1. Preparation of Deep Eutectic Solvents

Three DESs were selected for the extraction of bioactive compounds from samples: CC, CA, and TA. These compounds were used in the combinations CC:CA and CC:TA, based on preliminary tests. These DESs were prepared as described above [54,55,56]; the hydrogen bonding acceptor (HBA), CC, was mixed with the hydrogen bonding donor (HBD), CA or TA, in a molar ratio of 1:1. These solvents were prepared at 80 °C under agitation using a bath with agitation (Dubnoff bath, TE-053, Diadema, São Paulo, Brazil, 25 ± 5 rpm) until the formation of a transparent and homogeneous liquid was achieved (2–6 h).

The prepared DESs were mixed with water (20%) to reduce their viscosity and change their polarity in order to increase extraction efficiency by improving the solubility of the corresponding solvents [57], because the high viscosity of a DES can impair the extraction process and reduce the efficiency of mass transport [58,59]. The organic solvent used was ethanol (95.5%).

### 4.2. Materials

About 1 kg of fruits were collected in the rural area of Jussara–Goiás, Brazil (latitude: 15°51′31″ S, longitude: 50°52′9″ W), only from the harvest of 2023, which was sufficient for all analyses carried out in this study. The samples were cleaned, manually dusted, and stored in polyethylene containers, then frozen in a high-temperature freezer (Metalfrio, NF40S, São Paulo, Brazil) to −18 °C. The samples were kept under freezing conditions and transported to the laboratory of the State University of Maringá, Paraná, Brazil [60].

All reagents used in this study were of analytical grade.

### 4.3. Obtaining the Extract

Firstly, the fruits (pulp and peel) of mamacadela and cagaita were dehydrated at 45 °C in a greenhouse with air circulation. Immediately after this, 1 g of dry sample was weighed and 10 mL of solvent was added; the extraction took place at 30 °C or 60 °C temperature, and for 30 or 60 min in a bath (Dubnoff bath, TE-053, Diadema, São Paulo, Brazil) with agitation (35 ± 5 rpm) or without agitation; these parameters were defined in preliminary tests. After the above procedures, the extract was filtered and stored in amber glasses for further analysis.

### 4.4. Analysis of Color and Water Activity

The color of the samples was evaluated using a colorimeter (Konica Minolta, CR-400, Osaka, Japan), utilizing the three-parameter CIELab system. The equipment was previously calibrated with a standard white plate. The values of L* vary from 0 to 100, indicating the variation of color from black to white; axis a* shows the variation from red (+a*) to green (−a*), while b* shows from yellow (+b*) to blue (−b*). From these data, the whiteness index, Wi, (Equation (1) and the yellowing index, Yi, Equation (2) were calculated using the following equations [61,62].*Wi* = 100 − √(100 − *L**)^2^ + (*a**)^2^ + (*b**)^2^(1)*Yi = (142.86 ∗ b* ∗ (1/L*))*(2)

In addition, water activity (AW) was determined using the water activity meter (Aqua-Lab, 4TE, Logan, UT, USA).

### 4.5. Quantification of Bioactive Compounds

The samples used for this analysis were prepared according to the descriptions in Section 4.3.

#### 4.5.1. Total Phenolic Compounds

The analysis of phenolic compounds was performed according to the methodology described by Singleton and Rossi [63], and are expressed in mg GAE/100 g.

#### 4.5.2. DPPH Assay

The elimination of the DPPH^•^ (2,2-Diphenyl-1-Picrylhydrazyl) radical was analyzed according to the methodology of Thaipong et al. [64], and was determined by the colorimetric method at 515 nm; the readings were performed using a spectrophotometer (Bel UV-Vis, model Uv-m51, Piracicaba, Brazil). Trolox was used as a standard for the calibration curve; the results are expressed in µM Trolox Equivalent (TE)/g product [65], and the calibration curve was as follows: y = 0.1101x + 3.8612; R^2^ = 0.9979.

#### 4.5.3. ABTS Assay

The elimination of the ABTS^•+^ (2,2 Azino-bis (3-ethylbenzothiazoline-6-sulfonic acid) was conducted using the methodology of Rufino et al. [66]. The cationic radical ABTS^•+^ is formed by the reaction of 5 mL of aqueous solution of ABTS^•+^. The radical cation ABTS^•+^ was mixed with 88 μL of potassium persulphate (PP) (140 mol/L), allowing the reaction to take place at room temperature, in the dark, for 16 h. Before the test, the ABTS^•+^ solution was diluted in ethyl alcohol PA; the analysis was conducted in triplicate. An aliquot of 30 μL of extract was added to each tube, together with 3 mL of ABTS^•+^ reagent, already diluted. The samples were incubated for 6 min and protected from light at room temperature; the reading was made at a wavelength of 734 nm, using a spectrophotometer (Bel UV-Vis, model Uv-m51, Piracicaba, Brazil). The antioxidant capacity of the succulent extract was determined based on the standard curve of Trolox (2 mol/L), and is expressed in mg Trolox/g sample [65]. The calibration curve was as follows: y = −0.0003x + 0.6889; R^2^ = 0.9925.

#### 4.5.4. Ferric Reducing Antioxidant Power (FRAP)

The FRAP assay was performed according to the methodology of Benzie and Strain [67]. It was evaluated using TPTZ reagents (2,4,6-tris(2-pyridyl)-s-triazine), a 0.3 M acetate buffer, and 20 mM ferric chloride. The results relating to antioxidants are expressed in mg Trolox/100 g sample. Absorbance was verified at 595 nm, and the readings were performed using a spectrophotometer (Bel UV-Vis, model Uv-m51, Piracicaba, Brazil). The results are expressed in µM eq Fe_2_SO_4_/g product [65], and the calibration curve was as follows: y = 0.0016x − 0.0368; R^2^ = 0.9901.

#### 4.5.5. Flavonoids

For the total flavonoids, we used the method established by Alothman, Brat, and Karim [68], and the results are expressed in mg of quercetin/100 g of sample. The samples were analyzed in a colorimetric assay using aluminum chloride (AlCl_3_), sodium nitrite (NaNO_2_), and sodium hydroxide (NaOH). Absorbance was immediately verified by using a spectrophotometer (Bel UV-Vis, model Uv-m51, Piracicaba, Brazil) at 510 nm. The calibration curve was prepared using a standard quercetin solution, and the results are expressed in mg of quercetin equivalent (QE)/100 g of product [65]. The calibration curve was as follows: y = 0.0014x + 0.0697; R^2^ = 0.9904.

#### 4.5.6. Total Carotenoid Content

For the quantification of total carotenoid content, we followed the methodology described by Rodriguez-Amaya and Kimura [69]. The extracts obtained using ethanol (Section 4.3) were read at 450 nm using a spectrophotometer (Bel UV-Vis, model Uv-m51, Piracicaba, Brazil), and calculated according to Equation (3).(3)$$\mathrm{Content}\;\mathrm{of}\;\mathrm{total}\;\mathrm{carotenoids}\;(\mathrm{mg}/100\;g)=(\mathrm{Abs}*V*10,000)/(A_{1cm}^{1\%}*M*100)$$
where Abs = absorbance at which the extraction solution was read, which for this work was the wavelength of 450 nm for β-carotene; V = final volume of the extract (mL); $A_{1cm}^{1\%}$ = extinction coefficient or molar absorptivity coefficient of a pigment in a given specific solvent (2620 for β-carotene); M = mass of the sample used for analysis.

### 4.6. Electrospray Ionization with Triple Quadrupolar Mass Spectrometry (DI-ESI-MS)

Mass spectra were acquired following the methodology described by Nardino et al. [70], using a Premier XE (Águas, Milford, MA, USA) mass spectrometer with electrospray ionization (ESI) and triploquadrupole analyzer. Multiple reaction monitoring (MRM) analysis under the two different conditions (Table 5) was used, as reported in previous studies [71,72,73,74,75]. For the mobile phase, methanol and ammonium hydroxide (0.1%) were used. Initially, the extract was diluted in the mobile phase with an initial concentration of 1000 μg/mL; then, the solution was filtered with a 0.45 μm hydrophilic PVDF filter. For analysis, the filtered solution was diluted and injected into the equipment, with a final concentration of 1.0 μg/mL. The data were analyzed Software MassLynx version 4.4. The results are expressed in a.u.

### 4.7. Statistical Analysis of Data

The results were obtained in triplicate and were subjected to analysis of variance (ANOVA) and Tukey’s test for comparison of means (*p* < 0.05). Data were analyzed using the software Sisvar Version: 5.6 [76,77].

## 5. Conclusions

The fruits demonstrated satisfactory outcomes in terms of the extraction of bioactive compounds, with eutectic solvents exhibiting particular efficacy in the extraction of phenolic compounds, and ethanol proving more effective for the extraction of flavonoids. Concerning total carotenoids, mamacadela exhibited values twice as high as cagaita.

The efficacy of ethanol in extracting yellow pigments was substantiated through color analysis, particularly by the b* and Yi parameters, for both fruits. Furthermore, the use of the eutectic solvent CC:TA resulted in the identification by DI-ESI-MS of 29 and 27 compounds for mamacadela and cagaita, respectively, thereby demonstrating a greater affinity relative to the quantification and identification of phenolic compounds.

This study provides the industry with a comprehensive overview of extraction methods, thus assisting in the selection of the most suitable technique based on the desired compounds and the intended application of the extract, whether for use as a dye or for the extraction of high-value bioactive compounds.

## Figures and Tables

**Figure 1 plants-14-02596-f001:**
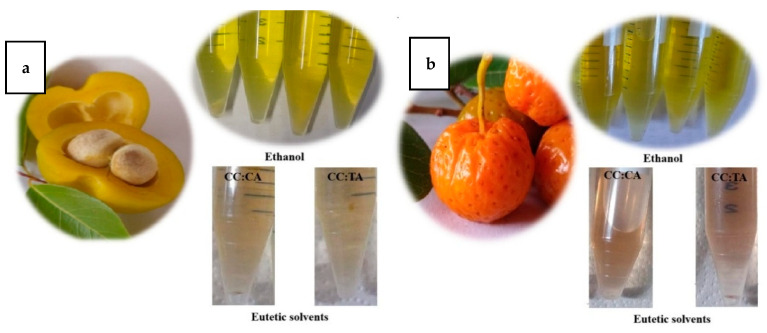
Fruits of the Cerrado: extracts obtained by eutectic solvents and ethanol for (**a**) cagaita and (**b**) mamacadela.

**Figure 2 plants-14-02596-f002:**
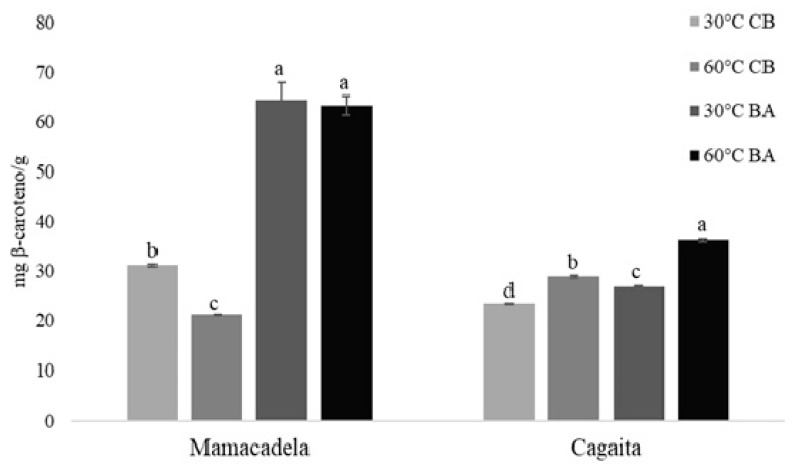
Total carotenoid contents for different extraction treatments, using ethanol as solvent extractor, for two different fruits of the Cerrado; eutectic solvents were not tested under this method. Error bars represent the standard deviations of the means, and different letters indicate significant differences (*p* < 0.05) between samples relative to the treatments for both fruits analyzed (cagaita and mamacadela).

**Table 1 plants-14-02596-t001:** Analysis of color and water activity for the extracts of cagaita and mamacadela.

**Mamacadela**						
**Extracts**	**L***	**a***	**b***	**Wi**	**Yi**	**AW**
**CC:TA 30 °C (CB)**	52.34 ± 0.92 ^aAB^	4.28 ± 0.13 ^bB^	19.68 ± 0.83 ^abDE^	48.25 ± 0.54 ^aA^	37.58 ± 0.97 ^dE^	0.50 ± 0.00 ^cD^
**CC:CA 30 °C (CB)**	51.01 ± 0.58 ^aAB^	4.70 ± 0.07 ^bB^	22.26 ± 0.25 ^abDE^	45.98 ^aA^ ± 0.41 ^aA^	43.65 ± 0.14 ^cdDE^	0.45 ± 0.07 ^cDE^
**Ethanol 30 °C (CB)**	29.88 ± 2.15 ^bC^	0.12 ± 2.46 ^cC^	13.39 ± 3.29 ^cE^	28.52 ± 1.55 ^cC^	44.53 ± 8.20 ^bcdCD^	0.97 ± 0.00 ^aA^
**Ethanol 60 °C (CB)**	29.42 ± 1.40 ^bC^	–1.78 ± 0.51 ^cC^	14.80 ± 1.78 ^cE^	27.84 ± 0.98 ^cC^	50.21 ± 3.61 ^abcC^	0.94 ± 0.02 ^aA^
**Ethanol 30 °C (BA)**	49.01 ± 7.79 ^aAB^	11.16 ± 0.86 ^aA^	28.13 ± 11.61 ^abCD^	39.65 ± 2.82 ^bB^	55.77 ± 16.69 ^abBC^	0.93 ± 0.02 ^aA^
**Ethanol 60 °C (BA)**	43.96 ± 5.49 ^aB^	9.55 ± 1.91 ^aA^	29.96 ± 5.23 ^aBCD^	35.49 ± 3.51 ^bB^	68.11 ± 7.72 ^aAB^	0.84 ± 0.00 ^bBC^
**Cagaita**						
**Extracts**	**L***	**a***	**b***	**Wi**	**Yi**	**AW**
**CC:TA 30 °C (CB)**	53.73 ± 3.05 ^aA^	–1.23± 0.09 ^aC^	22.50 ± 1.31 ^cDE^	48.50 ± 2.20 ^aA^	41.88 ± 0.06 ^cDE^	0.40 ± 0.06 ^aE^
**CC:CA 30 °C (CB)**	54.36 ± 2.81 ^aA^	–1.61 ± 0.08 ^aC^	20.77 ± 1.48 ^cDE^	49.79 ± 1.96 ^aA^	38.19 ± 0.76 ^cDE^	0.50 ± 0.00 ^aD^
**Ethanol 30 °C (CB)**	53.46 ± 0.80 ^aA^	–8.81 ± 0.41 ^bcD^	37.77 ± 2.34 ^bABC^	39.38 ± 0.87 ^bB^	70.62 ± 3.36 ^bAB^	0.93 ± 0.00 ^aA^
**Ethanol 60 °C (CB)**	52.88 ^a^ ± 0.99 ^aAB^	–8.08 ± 0.28 ^bcD^	40.74 ± 1.26 ^abA^	37.42 ± 0.22 ^bcB^	77.02 ± 1.04 ^aA^	0.91 ± 0.00 ^aAB^
**Ethanol 30 °C (BA)**	51.93 ± 0.95 ^aAB^	–8.90 ± 0.39 ^cD^	37.39 ± 1.90 ^bABC^	38.42 ± 0.44 ^bcB^	71.97 ± 2.36 ^bAB^	0.82 ± 0.00 ^aC^
**Ethanol 60 °C (BA)**	52.60 ± 0.34 ^aAB^	–8.32 ± 0.05 ^bD^	42.94 ± 0.43 ^aAB^	35.50 ± 0.18 ^cB^	81.63 ± 0.59 ^aA^	0.89 ± 0.00 ^aABC^

Mean values ± standard deviation (SD); *n* = 3 (triplicate); Equal superscript lowercase letters in the same column do not differ significantly (*p* < 0.05) among themselves relative to extracts for the same fruit (cagaita or mamacadela). Equal uppercase letters overwritten in the same column do not differ significantly (*p* < 0.05) among themselves relative to the treatments for both fruits analyzed (cagaita and mamacadela). CC:TA—Choline chloride with tartaric acid, ratio 1:1; CC:CA—Choline chloride with citric acid, ratio 1:1, with addition of 20% water; CB—Conventional bath; BA—Bath with agitation; WI—Whiteness index Wi; Yi—Yellowing index; AW—Water activity.

**Table 2 plants-14-02596-t002:** Antioxidant capacity of Cerrado fruits with different extraction solvents and quantification methodologies.

**Mamacadela**					
**Extracts**	**TPC**	**Flavonoids**	**ABTS** ^•+^	**FRAP**	**DPPH** ^•^
**CC:TA 30 °C (CB)**	144.21 ± 6.37 ^aA^	18.32 ± 7.03 ^cC^	26.01 ± 11.43 ^cC^	246.86 ± 4.98 ^aB^	349.90 ± 70.77 ^aCD^
**CC:CA 30 °C (CB)**	141.91 ± 21.15 ^aA^	22.03 ± 0.69 ^cC^	35.49 ± 0.00 ^cC^	102.92 ± 7.60 ^bCDE^	255.66 ± 59.83 ^aDE^
**Ethanol 30 °C (CB)**	1.40 ± 0.70 ^bE^	81.39 ± 6.82 ^bB^	2.39 ± 2.01 ^cC^	156.22 ± 76.88 ^bCDE^	ND
**Ethanol 60 °C (CB)**	8.30 ± 2.49 ^bDE^	143.53 ± 5.75 ^abAB^	138.73 ± 21.62 ^bBC^	123.04 ± 17.55 ^bBCDE^	ND
**Ethanol 30 °C (BA)**	5.74 ± 3.84 ^bE^	169.04 ± 5.11 ^aA^	109.10 ± 10.26 ^bC^	163.90 ± 54.69 ^abBCD^	ND
**Ethanol 60 °C (BA)**	11.65 ± 5.59 ^bDE^	135.18 ± 3.98 ^abAB^	190.88 ± 20.83 ^aBC^	153.32 ± 7.47 ^abCD^	ND
**Cagaita**					
**Extracts**	**TPC**	**Flavonoids**	**ABTS** ^•+^	**FRAP**	**DPPH** ^•^
**CC:TA 30 °C (CB)**	64.15 ± 9.38 ^bB^	20.64 ± 0.00 ^aC^	131.62 ± 19.80 ^cBC^	198.75 ± 10.72 ^bBC^	652.07 ± 7.12 ^aA^
**CC:CA 30 °C (CB)**	171.32 ± 21.52 ^aA^	21.57 ± 0.80 ^aC^	41.54 ^cC^ ± 32.26 ^cC^	53.56 ± 4.04 ^cE^	659.65 ± 35.70 ^aA^
**Ethanol 30 °C (CB)**	22.52 ± 2.24 ^cCDE^	12.29 ± 3.87 ^bcC^	1219.65 ± 448.70 ^abA^	510.68 ± 58.18 ^aA^	312.34 ± 2.48 ^cdCDE^
**Ethanol 60 °C (CB)**	43.82 ± 6.96 ^bcBC^	1.62 ± 0.80 ^dC^	1120.09 ± 322.21 ^abA^	81.77 ± 10.99 ^cDE^	493.57 ± 38.36 ^bB^
**Ethanol 30 °C (BA)**	23.49 ± 4.83 ^cCDE^	14.84 ± 1.06 ^bC^	1435.36 ± 70.60 ^aA^	508.40 ± 46.69 ^aA^	236.55 ± 38.86 ^dE^
**Ethanol 60 °C (BA)**	37.86 ± 8.03 ^bcBCD^	8.58 ± 0.80 ^cC^	597.41 ± 49.78 ^bcB^	63.51 ^cE^ ± 3.64 ^cE^	375.93 ± 17.01 ^cC^

TPC—total phenolic compounds, expressed in mg of gallic acid equivalent (GAE)/100 g); Flavonoids, expressed in mg of quercitin equivalent (EQ)/100 g); ABTS^•+^, FRAP, and DPPH^•^, (mg Trolox/100 g). Mean values ± standard deviation (SD); *n* = 3 (triple analysis); Equal superscript lowercase letters in the same column do not differ significantly (*p* < 0.05) among themselves relative to treatments for the same fruit (cagaita or mamacadela). Equal uppercase letters in the same column do not differ significantly (*p* < 0.05) between themselves regarding the treatments for both fruits analyzed (cagaita and mamacadela). CC:TA—Choline chloride with tartaric acid, ratio 1:1; CC:CA—Choline chloride with citric acid, ratio 1:1, with addition of 20% water; CB—Conventional bath; BA—Bath with agitation. ND—Not detected.

**Table 3 plants-14-02596-t003:** Phenolics identified in mamacadela extracts by MRM.

Bioactive Compound	[M − H]^−^ (*m*/*z*)	Fragmentation (*m*/*z*)	Mamacadela (a.u.)					
			CC:AT	CC:AC	Ethanol			
					30 °C CB	60 °C CB	30 °C BA	60 °C BA
Trans-cinnamic acid	147	103	243.47 ± 73.19	26.00 ± 5.56	8.97 ± 3.71	6.76 ± 1.31	6.30 ± 3.25	18.18 ± 17.72
*p*-Coumaric acid	163	119	141.00 ± 33.51	4.00 ± 1.73	6.87 ± 5.08	6.41 ± 2.69	12.41 ± 1.28	8.529 ± 4.37
Propoccecuic acid	153	109	ND	4.67 ± 2.08	4.14 ± 1.27	8.03 ± 5.60	7.98 ± 2.91	8.57 ± 5.41
Gallic acid	169	125	ND	16.33 ± 8.38	18.87 ± 12.35	26.89 ± 13.30	23.34 ± 11.05	34.24 ± 25.31
*p*-Hydroxybenzoic acid	137	93	115.00 ± 10.26	ND	10.98 ± 3.25	25.81 ± 14.3	15.63 ± 3.66	36.98 ± 32.98
Caffeic acid	179	135	157.00 ± 50.89	9.33 ± 2.08	4.98 ± 2.59	15.72 ± 2.66	5.34 ± 2.05	18.14 ± 7.25
Vanillic acid	167	108	137.50 ± 41.72	ND	ND	ND	ND	ND
Syringic acid	197	182	189.50 ± 10.60	ND	ND	ND	ND	ND
Catechin	289	245	267.00 ± 76.89	4.00 ± 1.73	ND	ND	ND	14.97 ± 6.98
Protocatechuic acid glucoside	315	153	234.00 ± 117.37	ND	ND	ND	ND	ND
Isoramnetine	315	300	307.50 ± 10.60	ND	ND	ND	ND	ND
Mircetina	317	151	234.00 ± 117.38	ND	ND	ND	ND	ND
Chlorogenic acid	353	135	244.00 ± 154.14	ND	3.79 ± 1.13	ND	ND	ND
3-O and 5-O Caffeoylquimic-acid	353.2	191	272.10 ± 114.69	27.00 ± 16.70	ND	ND	4.55 ± 1.34	ND
Chlorogenic acid	353.4	191	272.20 ± 114.83	15.00 ± 9.16	8.25 ± 3.99	ND	4.12 ± 1.92	ND
Rosmarinic acid	359	161	260.00 ± 198.34	ND	ND	ND	ND	ND
Salvianolic Acid A	493	295	394.00 ± 210.23	ND	ND	4.08 ± 0.97	ND	3.68 ± 1.18
Salvianolic Acid H	537	339	438.00 ± 120.56	4.00 ± 1.00	ND	ND	ND	ND
Eriordictyol-O-Rutinoside	595	287	441.00 ± 217.79	7.00 ± 5.29	ND	3.85 ± 1.46	ND	3.42 ± 0.58
Salvianolic Acid B	717	393	555.00 ± 229.10	7.67 ± 4.16	ND	ND	ND	ND
Salvianolic acid E	717	537	627.00 ± 127.27	3.00 ± 0.00	ND	ND	ND	ND
Kaempferol	258	161	209.50 ± 68.59	ND	ND	3.67 ± 1.15	3.67 ± 0.57	ND
Luteolina	285	151	218.00 ± 94.75	ND	ND	ND	ND	4.96 ± 1.51
5,7,3′,41-Flavan-3-OL (quercetin)	301	151	226.00 ± 106.06	9.00 ± 5.57	8.13 ± 5.37	10.15 ± 4.19	5.32 ± 1.62	6.76 ± 3.40
Hiesperetina	301	286	293.50 ± 10.60	ND	ND	ND	ND	ND
5,7,3′,41-Flavan-3-OL (quercetin)	301	299	300.00 ± 1.41	ND	ND	ND	ND	ND
Cyanidine-3-O-arabinoside	418	287	352.50 ± 92.63	60.33 ± 32.12	71.47 ± 4.54	23.77 ± 13.47	30.32 ± 18.90	31.41 ± 6.05
Cyanidine-3-O-Glucoside	448	287	367.50 ± 113.84	5.33 ± 2.51	ND	ND	ND	4.33 ± 1.15
Luteolin-7-O-Glucuronide	461	285	373.00 ± 124.45	35.33 ± 11.54	4.44 ± 1.33	5.67 ± 1.85	6.54 ± 2.96	ND
Kaempferol-3-malonihexoside	533	285	409.00 ± 175.36	77.00 ± 32.23	ND	ND	8.50 ± 3.70	5.00 ± 1.85
Naringerin-O-Rutinoside	579	271	425.00 ± 217.78	3.33 ± 0.58	ND	ND	ND	ND
Hesperetin-O-Rutinoside	609	301	ND	3.00 ± 0.00	ND	ND	ND	ND
Catechin	289	245	ND	ND	ND	4.38 ± 0.69	ND	7.55 ± 6.90
Narigenina	217	151	ND	ND	ND	6.06 ± 5.30	ND	ND
Apigenin-O-Rutinoside	577	269	ND	ND	ND	ND	6.46 ± 0.80	ND
Diosmin	607	299	ND	ND	4.00 ± 0.89	ND	ND	ND
Cyanidine-3,5,O-Dihexiside	610	287	ND	ND	ND	4.53 ± 2.20	ND	4.72 ± 2.13
Cyanidine-3,5,O-Dihexiside	610	448	ND	ND	8.97 ± 3.71	6.76 ± 1.31	6.30 ± 3.25	18.18 ± 17.72

Mean values of the peak area ± standard deviation (SD); *n* = 3 (triple analysis). CC:TA—Choline chloride with tartaric acid, ratio 1:1; CC:AC—Choline chloride with citric acid, ratio 1:1, with addition of 20% water; CB—Conventional bath; BA—Bath with agitation. ND—Not detected; a.u—Arbitrary unit.

**Table 4 plants-14-02596-t004:** Phenolics identified in cagaita extracts by MRM.

Bioactive Compound	[M − H]^−^ (*m*/*z*)	Fragmentation (*m*/*z*)	Cagaita (a.u.)					
			CC:AT	CC:AC	Ethanol			
					30 °C CB	60 °C CB	30 °C BA	60 °C BA
Trans-cinnamic acid	147	103	107.00 ± 21.51	31.33 ± 20.13	10.10 ± 0.88	8.59 ± 4.24	8.64 ± 4.74	6.69 ± 2.66
*p*-Coumaric acid	163	119	10.00 ± 7.81	7.67 ± 7.23	6.86 ± 4.16	5.07 ± 1.86	4.95 ± 1.69	4.84 ± 1.69
Propocaceic acid	153	109	4.33 ± 0.58	8.67 ± 3.78	4.60 ± 0.62	ND	4.16 ± 1.14	4.25 ± 1.57
Gallic acid	169	125	19.67 ± 9.01	7.00 ± 2.64	27.08 ± 8.87	34.70 ± 15.56	19.26 ± 9.81	15.43 ± 8.40
*p*-Hydroxybenzoic acid	137	93	54.00 ± 30.31	25.67 ± 8.38	20.52 ± 2.25	17.96 ± 10.64	12.50 ± 4.22	21.79 ± 6.46
Caffeic acid	179	135	6.33 ± 3.05	4.33 ± 1.52	10.13 ± 4.20	9.74 ± 5.43	5.82 ± 3.29	13.50 ± 5.38
Ferrulic acid	193	134	5.00 ± 3.46	ND	ND	ND	5.33 ± 2.51	ND
Siringic acid	197	182	5.67 ± 3.78	5.33 ± 2.51	4.38 ± 2.37	ND	ND	ND
Catechin	289	245	5.33 ± 4.04	ND	4.43 ± 1.63	5.64 ± 1.47	8.10 ± 5.73	ND
Protocatechuic acid glucoside	315	153	ND	8.67 ± 6.42	ND	ND	4.26 ± 1.44	ND
Isoramnetine	315	300	4.00 ± 1.73	ND	ND	ND	ND	ND
Mircetina	317	151	7.67 ± 3.51	6.33 ± 0.57	ND	4.27 ± 2.20	ND	4.00 ± 0.18
Chlorogenic acid	353	135	9.33 ± 5.03	6.67 ± 3.21	5.13 ± 3.16	ND	ND	4.85 ± 0.41
3-O and 5-O Caffeoylquimic-acid	353.2	191	ND	57.33 ± 39.92	4.30 ± 1.14	ND	9.14 ± 2.61	4.59 ± 1.66
Chlorogenic acid	353.4	191	ND	11.67 ± 10.69	3.82 ± 0.79	5.84 ± 2.45	ND	6.94 ± 3.74
Rosmarinic acid	359	161	ND	ND	3.63 ± 0.52	ND	ND	ND
Salvianolic Acid A	493	295	ND	ND	3.25 ± 0.35	ND	ND	4.00 ± 0.8
Salvianolic Acid H	537	339	4.00 ± 1.00	5.00 ± 2.00	ND	ND	ND	ND
Salvianolic Acid B	717	393	5.33 ± 4.04	14.33 ± 10.06	ND	ND	ND	ND
Salvianolic acid E	717	537	3.67 ± 1.15	3.67 ± 1.15	6.35 ± 2.68	ND	ND	ND
Kaempferol	258	161	ND	ND	ND	3.67 ± 1.15	ND	ND
Luteolina	285	151	6.33 ± 5.77	8.67 ± 7.23	4.93 ± 2.45	7.35 ± 5.76	4.82 ± 1.56	7.71 ± 3.23
Luteolina	285	259	3.33 ± 0.58	3.33 ± 0.57	ND	ND	ND	ND
5,7,3′,41-Flavan-3-OL (quercetin)	301	151	ND	ND	8.40 ± 5.01	15.34 ± 8.38	ND	12.01 ± 6.10
5,7,3′,41-Flavan-3-OL (quercetin)	301	299	ND	ND	ND	4.03 ± 1.52	ND	ND
Cyanidine-3-O-Arabinoside	418	287	71.67 ± 40.70	89.33 ± 7.63	48.21 ± 25.04	63.94 ± 17.81	101.09 ± 54.49	73.06 ± 16.56
Luteolin-7-O-glucuronide	461	285	3.33 ± 0.58	5.00 ± 2.00	6.49 ± 1.79	8.68 ± 4.83	7.86 ± 4.70	ND
Kaempferol-3-malonihexoside	533	285	27.33 ± 9.81	ND	5.38 ± 3.99	4.07 ± 0.93	ND	ND
Naringerin-O-Rutinoside	579	271	16.33 ± 10.01	78.67 ± 10.21	ND	ND	ND	21.19 ± 12.90
Hesperetin-O-Rutinoside	609	301	ND	ND	ND	ND	ND	6.00 ± 2.64
Quercitin-3-O-Rutinoside (RUTIN)	609	300	3.00 ± 0.00	ND	ND	ND	ND	ND
Luteolin-O-Diglucuronide	637	285	4.00 ± 1.73	ND	ND	ND	ND	ND
Catechin	289	245	5.33 ± 4.04	ND	8.97 ± 0.94	ND	8.60 ± 4.29	ND
Narigenina	217	151	4.67 ± 2.08	15.00 ± 6.08	4.11 ± 1.93	6.57 ± 3.73	ND	7.52 ± 5.44
Myrcetin-O-Glucoside	479	317	12.33 ± 7.50	6.33 ± 4.04	7.61 ± 3.54	ND	ND	ND
Apigenin-O-Rutinoside	577	269	5.33 ± 4.04	3.33 ± 0.57	ND	4.80 ± 0.67	3.30 ± 0.37	5.086 ± 1.23
Kaenferol	285	145	ND	ND	6.16 ± 3.14	ND	ND	ND
Diosmin	607	299	ND	ND	4.00 ± 0.89	ND	ND	5.30 ± 3.10
Eriordictyol-O-Hexoside	499	287	ND	ND	ND	ND	3.96 ± 0.80	ND
Kaempferol-Hexoside	447	285	ND	ND	ND	ND	ND	4.86 ± 2.44
Cyanidine-3,5,O-Dihexisid	610	287	ND	ND	ND	4.95 ± 2.92	ND	4.19 ± 1.42
Cyanidine-3,5,O-Dihexisid	610	448	ND	ND	10.10 ± 0.88	8.59 ± 4.24	8.64 ± 4.74	6.69 ± 2.66

Mean values of the peak area ± standard deviation (SD); *n* = 3 (triple analysis). CC:TA—Choline chloride with tartaric acid, ratio 1:1; CC:CA—Choline chloride with citric acid, ratio 1:1, with addition of 20% water; CB—Conventional bath; BA—Bath with agitation. ND—Not detected; a.u—Arbitrary unit.

**Table 5 plants-14-02596-t005:** Mass spectrometry (MS/MS) analysis conditions utilizing the MRM method.

Component	Condition 1	Condition 2
Desolvation gas temperature/°C	200	250
Source gas temperature/°C	110	110
Ionization mode	Negative	Negative
Capillary voltage/kV	2.0	2.5
Cone voltage/V	20.0	40.0
Collision energy/V	15.0	30.0
Collision gas pressure/mmHg	3.5 × 10^−3^	3.5 × 10^−3^

MRM: multiple reaction monitoring.

## Data Availability

The datasets supporting the conclusions of this article are included within the manuscript.

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
