# Peer review of "Enhanced Recovery of Bioactive Compounds from Cagaita and Mamacadela Fruits Using Natural Deep Eutectic Solvents (NADES) and Ethanol: A Comparative Study"

_plants, 2025, doi:10.3390/plants14162596_

Round 1

Reviewer 1 Report

Comments and Suggestions for Authors

The manuscript title “Enhanced Recovery of Bioactive Compounds from Cagaita and Mamacadela Fruits Using Natural Deep Eutectic Solvents (NADES) and Ethanol: A Comparative Study” is conducted well and have scientific worth, however, it needs significant revisions and explanation.

Comments for authors are as follows:

  • Line 61-65: “Studies on native fruits are an alternative for preservation in view that they would provide a sustainable alternative with the commercialization of native fruits instead of products such as soy, another advantage is that these fruits are found in the native flora of the region, which consequently the time of their commercialization would result in the conservation of the biome.” The authors have written several very long sentences, hard to understand, I suggest authors split into small sentences, it will be easy to understand.
  • Also, in line 61 authors mentioned “Studies on native fruits” which studies?? Authors didn’t cite any study here. Why?
  • Line 68-71: “The choice of extraction solvent plays a crucial role in the extraction of natural compounds due to the effects of different solvents on major chemical components and their biological activities, including antioxidants, as each solvent has molecular weight, different yield, chemical status, and bioactivity” same problem as I mentioned in comment 1. The authors have written several very long sentences, hard to understand, I suggest authors split into small sentences, it will be easy to understand.
  • Figure 1 captions are very blur. Upload a clear captions figure.
  • Figure 2: each column has a bar, what does the bar stand for?
  • In table 3: the Trans-cinnamic acid, p-Coumaric acid, p-Hydroxybenzoic acid, Caffeic acid etc… the a ± standard deviation (SD) values in CC:AT column are same ±31.11. why? Moreover, some SD values are very big such as for Rosmarinic acid the SD value is ±140.00 mentioned by authors? Same with Salvianolic Acid A and Salvianolic Acid H. How? And some have even Naringerin-O-Rutinoside ±217.78 SD value. How? Authors are requested to Recheck the data.
  • Line 357: “The analysis of phenolic compounds was performed according to the methodology described by Singleton & Rossi [61], expressed in mg GAE /100 g.” just writing this and mentioning the reference is not enough, please specify the preparation of samples, how much samples was used fresh/dried? On which apparatus the phenolic were measured? Same with all other metabolic and antioxidant determinants.
  • ABTS is radical cation, it should be written as (ABTS·+) in whole manuscript.
  • Section 4.5. explain all methods with necessary details and the concentration standards should be mentioned.
  • Line 390: authors mentioned ANOVA. Where is the ANOVA table?
  • All the abbreviations mentioned in the text should be explained with their full names when they appear for the first time in the text, after that you can use abbreviations.

Comments on the Quality of English Language

some sentences are very long, and needs to be revised.

Author Response

Comments and Suggestions for Authors

The manuscript title “Enhanced Recovery of Bioactive Compounds from Cagaita and Mamacadela Fruits Using Natural Deep Eutectic Solvents (NADES) and Ethanol: A Comparative Study” is conducted well and have scientific worth, however, it needs significant revisions and explanation.

Comments for authors are as follows:

  • Line 61-65: “Studies on native fruits are an alternative for preservation in view that they would provide a sustainable alternative with the commercialization of native fruits instead of products such as soy, another advantage is that these fruits are found in the native flora of the region, which consequently the time of their commercialization would result in the conservation of the biome.” The authors have written several very long sentences, hard to understand, I suggest the authors split into small sentences, it will be easy to understand.

Thank you for your comment, reviewer #1. The text has been reviewed. Please check lines 61-64.

  • Also, in line 61 authors mentioned “Studies on native fruits” which studies?? Authors didn’t cite any study here. Why?

Thank you for your comment, reviewer #1. The text has been reviewed. Please check lines 61-64.

  • Line 68-71: “The choice of extraction solvent plays a crucial role in the extraction of natural compounds due to the effects of different solvents on major chemical components and their biological activities, including antioxidants, as each solvent has molecular weight, different yield, chemical status, and bioactivity” same problem as I mentioned in comment 1. The authors have written several very long sentences, hard to understand, I suggest authors split into small sentences, it will be easy to understand.

Thank you for your comment, reviewer #1. The text has been reviewed. Please check lines 66-70.

  • Figure 1 captions are very blur. Upload a clear captions figure.

Thank you for your comment, Reviewer #1. Figure 1 was sent individually to provide better resolution.

  • Figure 2: each column has a bar, what does the bar stand for?

Thank you for your comment, reviewer #1. The caption for Figure 2 has been revised to provide better interpretation. 

  • In table 3: the Trans-cinnamic acid, p-Coumaric acid, p-Hydroxybenzoic acid, Caffeic acid etc… the a ± standard deviation (SD) values in CC:AT column are same ±31.11. why? Moreover, some SD values are very big such as for Rosmarinic acid the SD value is ±140.00 mentioned by authors? Same with Salvianolic Acid A and Salvianolic Acid H. How? And some have even Naringerin-O-Rutinoside ±217.78 SD value. How? Authors are requested to Recheck the data.

Thank you for your comment, reviewer #1. In fact, the high values of standard deviation observed for some compounds such as rosmarinic acid, salvianolic acid A, salvianolic acid H and Naringenine-O-rutinoside, are related to the variability in the area of peaks obtained by mass spectrometry, which can be influenced by several factors, including the complexity of the matrix and the intensity of the detected signal. We would like to clarify that the data presented refer to the relative presence of compounds in the samples based on the areas of the peaks of the extracted ions and not on an absolute quantification with analytical standards. Therefore, the standard deviations reflect the natural variation of the detector response between biological and/or technical replicates, not necessarily inconsistencies in the presence of the compound. We reviewed the data and confirmed that the values presented correspond to the variability observed experimentally. We believe that, even with this variability, the data are valid to indicate the presence of compounds in different samples. The same standard deviation values for different compounds were revised and corrected.

  • Line 357: “The analysis of phenolic compounds was performed according to the methodology described by Singleton & Rossi [61], expressed in mg GAE /100 g.” just writing this and mentioning the reference is not enough, please specify the preparation of samples, how much samples was used fresh/dried? On which apparatus the phenolic were measured? Same with all other metabolic and antioxidant determinants.

Thank you for your comment, reviewer #1. We appreciate the pertinent remark. We agreed that the methodological description was succinct and therefore revised this section to include detailed information on sample preparation, amount used (fresh/dry sample mass), as well as the equipment and analytical conditions used for the determination of phenolic compounds and other metabolic parameters and antioxidants. This information was added to the section of Materials and Methods, to ensure greater clarity and reproducibility of experiments performed.

  • ABTS is radical cation, it should be written as (ABTS·+) in whole manuscript.

Thank you for your comment, reviewer #1. The description of the radicals (ABTS•+ and DHHP) has been revised throughout the text.

  • Section 4.5. explain all methods with necessary details and the concentration standards should be mentioned.

Thank you for your comment, reviewer #1. The requested information was duly added to the Materials and Methods section, including details about sample preparation, amount used (fresh/dry sample mass), equipment used, and analytical conditions employed. In addition, the concentrations and other parameters related to metabolic and antioxidant determinations were also specified in the text.

  • Line 390: authors mentioned ANOVA. Where is the ANOVA table?

Thank you for your comment, reviewer #1. ANOVA was performed to verify differences between the groups, with significant results. Since the focus was to identify which groups differed, we chose to present only the Tukey test. The ANOVA table can be included if necessary.

  • All the abbreviations mentioned in the text should be explained with their full names when they appear for the first time in the text, after that you can use abbreviations.

Thank you for your comment, reviewer #1. All abbreviations used in the manuscript were reviewed, and full names were included the first time each appears in the text, as suggested. After this first mention, we use only the abbreviations to keep the fluidity of the reading.

Reviewer 2 Report

Comments and Suggestions for Authors

Studies that describe and compare the best extraction solvents allow the industry to select them according to their ultimate goal and the compounds of interest. The manuscript “Enhanced Recovery of Bioactive Compounds from Cagaita and Mamacadela Fruits Using Natural Deep Eutectic Solvents (NADES) and Ethanol: A Comparative Study” aims to 1) extract bioactive compounds from two Cerrado fruit species (Mamacadela and Cagaita) under different conditions and solvents (eutectic and organic), 2) evaluate the color, water activity and quantify the bioactive compounds, 3) identify by mass spectrometry (DI-ESI-MS extract of the compounds present in the different extracts).

Comment 1. In the results, e.g. starting from Table 1 and other tables and figures, instead of -, use – . It is not usual to use small letters that indicate the statistical significance of differences like this 144.21aA±6.37, but like this 144.21±6.37aA.

Comment 2. All figures are of very poor quality. Improve resolution, format, etc.

Comment 3. In part 4.2. Material, it is stated that the fruit as plant material for testing was collected in the harvest of 2023. Is it sufficient for such an examination to be based on material from only one year of harvest? Or from several years? It should be explained.

Comment 4. The CIELab system and colorimeter (Konica Minolta, CR-400, Osaka, Japan) were used for color analysis. Is the device calibrated before measuring the color and how?

Comment 5. In Line 354 this is not clear, ... whiteness index (Wi, Eq. (1)) and yellowing index (Yi, Eq. (2)) ..., I suggest ... whiteness index, Wi (Eq. (1) and yellowing index, Yi, (Eq. (2) ... Also, equation (5) is not clearly written. Please improve.

Author Response

#Reviewer 2

Studies that describe and compare the best extraction solvents allow the industry to select them according to their ultimate goal and the compounds of interest. The manuscript “Enhanced Recovery of Bioactive Compounds from Cagaita and Mamacadela Fruits Using Natural Deep Eutectic Solvents (NADES) and Ethanol: A Comparative Study” aims to 1) extract bioactive compounds from two Cerrado fruit species (Mamacadela and Cagaita) under different conditions and solvents (eutectic and organic), 2) evaluate the color, water activity and quantify the bioactive compounds, 3) identify by mass spectrometry (DI-ESI-MS extract of the compounds present in the different extracts).

Comment 1. In the results, e.g. starting from Table 1 and other tables and figures, instead of -, use – . It is not usual to use small letters that indicate the statistical significance of differences like this 144.21aA±6.37, but like this 144.21±6.37aA.

Thank you for your comment, reviewer #2. The data in Table 1 have been revised as suggested. Tukey's significant difference expressions have been revised.

Comment 2. All figures are of very poor quality. Improve resolution, format, etc.

Thank you for your comment, reviewer #2. The figures were sent individually to provide better resolution.

Comment 3. In part 4.2. Material, it is stated that the fruit as plant material for testing was collected in the harvest of 2023. Is it sufficient for such an examination to be based on material from only one year of harvest? Or from several years? It should be explained.

Thank you for your comment, reviewer #1. We recognize that studies with collections carried out in different harvests can provide a more comprehensive view of the seasonal or environmental variations that influence the chemical composition of fruits. However, in this study, we chose to use samples collected in the 2023 harvest to perform an initial exploratory analysis of the chemical profile and antioxidant activity of the species in question. We understand that this is a limitation and highlight this point in the manuscript, indicating that future studies will include samples from different years and regions for a more robust assessment of the variability of bioactive compounds. This initial approach, however, provides important and unprecedented data on the potential of the species studied. In time, we mention that other studies are being carried out with these fruits by our research group

Comment 4. The CIELab system and colorimeter (Konica Minolta, CR-400, Osaka, Japan) were used for color analysis. Is the device calibrated before measuring the color and how?

Thank you for your comment, reviewer #2. The text has been reviewed. Please check lines 347-352.

Comment 5. In Line 354 this is not clear, ... whiteness index (Wi, Eq. (1)) and yellowing index (Yi, Eq. (2)) ..., I suggest ... whiteness index, Wi (Eq. (1) and yellowing index, Yi, (Eq. (2) ... Also, equation (5) is not clearly written. Please improve.

Thank you for your comment, reviewer #2. The text has been reviewed. Please check line 352.

On the authors’ behalf, thank you in advance. 

Best regards,

*Correspondence: gsmadrona@uem.br

Round 2

Reviewer 1 Report

Comments and Suggestions for Authors

the authors did sufficient revisions.